∂ | **Open Peer Review** | Virology | Research Article

# HERV6196 as an enhancer with oncogenic potential in rectal cancer

Yi-Xiu Gan,[1,2] Xin Jiang,[3,4] Zhi-Yu Wang,[1] Yi-Lin Yu,[1] Ling-Dong Shao,[1] Jian-Min Wang,[3,4] Jun-Xin Wu[1]

**ABSTRACT** Colorectal cancer (CRC) is among the most prevalent malignancies. However, the regulatory networks involved in tumor occurrence and development are still poorly understood. Human endogenous retroviruses (HERVs), a class of transposable elements, have been implicated in the development and progression of various human cancers. This study presents the first comprehensive locus-specific profiling of the expression of HERV gene transcripts in rectal cancer, revealing significantly dysregulated HERVs. Analysis of data from three Gene Expression Omnibus data sets revealed 25 upregulated HERVs and 7 downregulated HERVs. Dysregulation of HERV6196, a type of HERVH, was validated through reverse transcription quantitative PCR and droplet digital PCR in cells and tissues. Additionally, HERV6196 promoted the proliferation, inhibited the apoptosis, enhanced the colony formation ability, and enhanced the migration capability of rectal cancer cells. Moreover, HERV6196 functioned as an enhancer, promoting the expression of neighboring genes and the development of CRC. In summary, the present results revealed that HERV6196 is involved in the pathogenesis of rectal cancer, indicating the potential contribution of dysregulated HERVs to the development and progression of CRC through gene expression modulation.

**IMPORTANCE** The role of human endogenous retroviruses (HERVs) in colorectal cancer (CRC) remains insufficiently understood. The present study revealed aberrant expression of HERV gene transcripts in cancerous tissues compared with non-cancerous tissues. HERV6196 contributes to CRC progression by regulating the expression of neighboring genes. These findings suggest that HERVs may serve as enhancers and regulate oncogenic gene expression, providing new insights for rewiring transcriptional regulatory networks in CRC pathogenesis.

**KEYWORDS** colorectal cancer, endogenous retrovirus, dysregulated expression

Colorectal cancer (CRC) is a leading malignant neoplasm worldwide. According to the 2020 data of the World Health Organization, CRC is the third most common cancer in terms of incidence and the second most common in terms of mortality (1). The majority of CRC cases arise from the gradual progression of precancerous lesions (2). Early diagnosis and treatment of CRC are best achieved through the use of tumor markers, imaging techniques, and genetic testing, which have the potential to reduce disease morbidity and mortality (3–5). However, current tests cannot rapidly and accurately detect CRC (6, 7). Thus, there is an urgent need to identify novel markers that can predict the occurrence of CRC.

Human endogenous retroviruses (HERVs) originated 30 million years ago when exogenous retroviruses infected human ancestors. Through continuous evolution, HERVs have integrated into the human genome and become a part of it. HERVs make up approximately 8%–9% of the human genome (8–10). A typical HERV structure consists of LTR-gag-pol-env-LTR, where the long terminal repeats (LTRs) serve as highly active

**Peer Reviewer** Michal Izydorczyk, Oxford Brookes University, Oxford, United Kingdom

Address correspondence to Jian-Min Wang, wangjm8605@163.com, or Jun-Xin Wu, junxinwufj@aliyun.com.

Yi-Xiu Gan and Xin Jiang contributed equally to this article. Author order was determined alphabetically by last name.

The authors declare no conflict of interest.

See the funding table on p. 15.

regulatory elements for transcription (11). HERVs are extensively distributed throughout the human body. Through genome-wide association studies, researchers have systematically identified 13,889 HERVs expressed in various normal tissues, revealing specific expression patterns associated with body part, sex, race, and age (12). Furthermore, studies have demonstrated that HERVs are reactivated during aging, leading to cellular aging and inflammation through the activation of the cGAS/STING innate immune pathway (13). In cancer research, HERVs may function as potential enhancers of neighboring oncogene expression, thereby contributing to the development of leukemia (14). In CRC, the activation of the LTR10 enhancer of the HERV family has been shown to contribute to transcriptional dysregulation in response to oncogenic signaling (15). However, the carcinogenic characteristics of individual HERV loci have not been comprehensively studied in CRC. The current knowledge on the expression of HERV gene transcripts and its correlation with rectal cancer progression is limited. Investigating HERV transcriptional patterns in rectal cancer could reveal novel virus-related cancer biomarkers and guide the development of more effective treatment strategies.

The repetitive nature of HERVs presents challenges in accurately measuring their expression across the genome via sequencing, hindering comprehensive research and progress (16). ERVmap is a pipeline designed for the quantification of proviral endogenous retroviruses in RNA-seq data, which encompasses all known proviral HERV sequences and employs algorithms with stringent filtering criteria to increase the reliability of HERV site identification. The application of ERVmap across a range of diseases and experimental conditions holds promise for identifying novel disease markers (17).

In this study, we employed ERVmap to analyze raw RNA-seq data from rectal cancer patients extracted from the NCBI Sequence Read Archive (SRA) database, which revealed abnormal expression of HERV gene transcripts in rectal cancer tissues compared with adjacent tissues, and these findings were validated through reverse transcription quantitative PCR (RT-qPCR) and droplet digital PCR (ddPCR). In addition, the present results indicated that HERV6196 promotes the proliferation and migration of rectal cancer cells while inhibiting their apoptosis. Finally, the present study suggested that HERV6196 contributes to the initiation and progression of rectal cancer by influencing neighboring oncogenes.

## RESULTS

### Differential expression of HERVs in rectal cancer tissues and adjacent non-cancerous tissues

In this study, we obtained raw RNA sequencing data from three Gene Expression Omnibus data sets, namely, GSE50760, GSE104836, and GSE142279. We subsequently quantitatively analyzed the RNA-seq data from rectal cancer and adjacent normal tissues via the ERVmap tool, referencing a gene locus-specific expression profile derived from a database of 3,220 HERVs. We analyzed the differentially expressed HERVs via the DESeq2 package (Table S1). In the GSE50760 data set, 107 HERVs exhibited differential expression, with 79 upregulated HERVs and 28 downregulated HERVs (Fig. 1A). Similarly, the GSE104836 data set revealed 86 upregulated HERVs and 47 downregulated HERVs. In the GSE142279 data set, 239 HERVs were differentially expressed, with 123 upregulated HERVs and 116 downregulated HERVs. To further investigate HERVs related to rectal cancer, we conducted an analysis of the intersection of the GSE50760, GSE104836, and GSE142279 data sets, which revealed 25 significantly upregulated HERVs (HEV6196, HERV4098, HERV6114, HERV1066, HERV4849, HERV1412, HERV557, HERV915, HERV3452, HERV1074, HERV526, HERV5305, HERV2916, HERV4270, ERVH13q33.3, HERV1475, HERV2003, HERV3730, HERV858, HERV3732, HERV4627, HERV4825, HERV5290, HERV2415, and HERV4719) (Fig. 1B) and seven significantly downregulated HERVs (HERV4152, HERV6007, HERV559, HERV1520, HERV2674, HERV6119, and HERV4457; Fig. 1C). Heatmaps were generated to visualize the differential expression of HERVs (Fig. 1D).

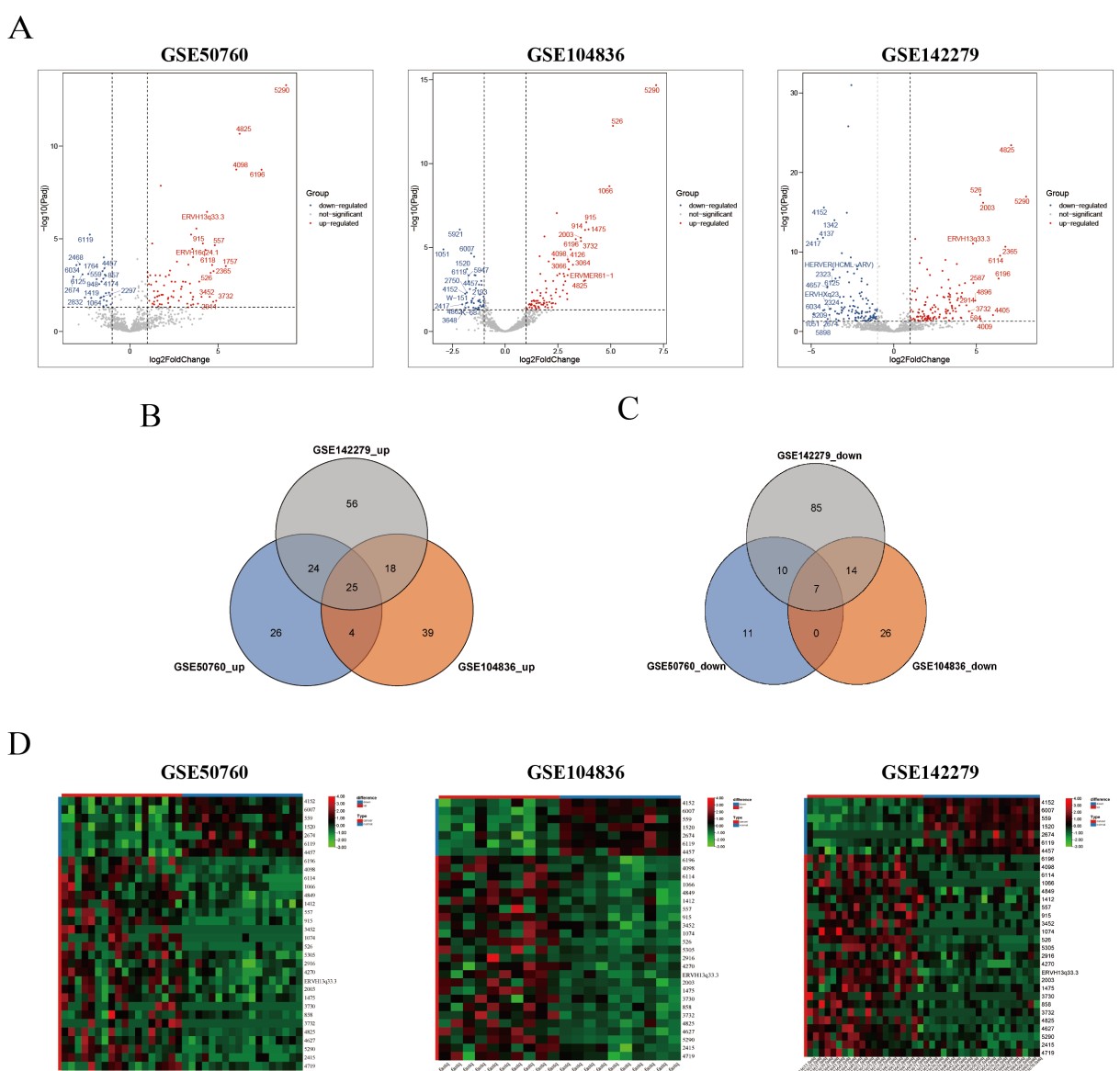

**FIG 1** Characterization of differentially expressed HERVs in CRC tissues. (A) Volcano plot of differentially expressed HERVs in the GSE50760, GSE104836, and GSE142279 data sets. Red represented the fifteen most highly upregulated HERVs, and blue represented the 15 most highly downregulated HERVs. The horizontal axis represented fold change, and the vertical axis represented statistical significance. (B) Venn diagram of upregulated HERVs across three data sets. (C) Venn diagram of downregulated HERVs across three data sets. (D) Heatmap plot of differentially expressed HERVs in the GSE50760, GSE104836, and GSE142279 data sets. Red on the horizontal axis represented tumor samples, and blue on the horizontal axis represented normal samples. Red on the vertical axis represented upregulated HERVs, and blue on the vertical axis represented downregulated HERVs.

## Verification of the differential expression of HERVs in rectal cell lines and tissues

To further validate the differential expression of HERVs at the cellular level, we randomly selected HERVs with a large |log2FoldChange| for RT-qPCR verification. HERV6196 expression was greater in three distinct rectal cancer cell lines (HT29, HCT116, and SW480) than in HCoEpiC (Fig. 2A). In contrast, compared with HCoEpiC, HERV6119 was expressed at lower levels in three distinct rectal cell lines (Fig. 2B). These findings were consistent with the results of the bioinformatics analysis conducted via ERVmap. Additionally, we collected 18 cancerous and adjacent tissues from patients with CRC and

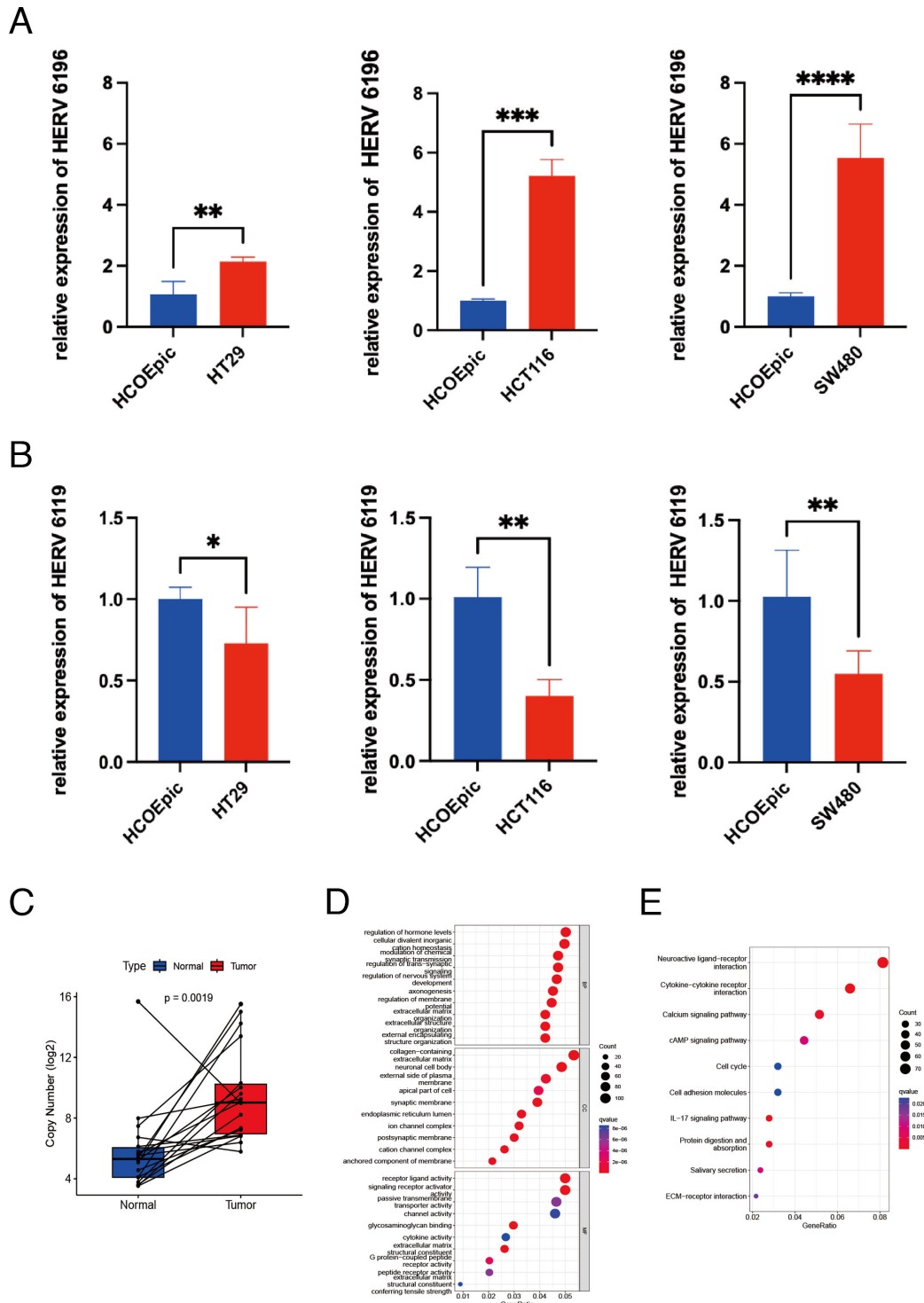

**FIG 2** Validation of the expression of HERV gene transcripts and pathway enrichment analysis of genes associated with HERV6196. (A) RT-qPCR analysis revealed that the expression of HERV6196 gene transcripts was significantly elevated in HT29, HCT116, and SW480 cells compared to that in HCoEpiC cells (normal colonic epithelial cells). (B) RT-qPCR analysis revealed that the expression of HERV6119 gene transcripts was significantly reduced in HT29, HCT116, and SW480 cells compared to that in HCoEpiC cells. Red represented CRC cells, and blue represented normal colonic epithelial cells. Significant values were noted as *$P < 0.05$, **$P < 0.01$, ***$P < 0.001$, and ****$P < 0.0001$. (C) The copy number of HERV6196 was significantly higher in paired CRC tissues compared to adjacent normal tissues. Red represented CRC tissues, and blue represented adjacent normal tissues. (D) Gene Ontology (GO) pathway enrichment analysis of genes associated with HERV6196. BP, biological progress; CC, cellular component; MF, molecular function. (E) Kyoto Encyclopedia of Genes and Genomes (KEGG) pathway enrichment analysis of genes associated with HERV6196.

conducted ddPCR experiments. The results expression of HERV6196 was greater in rectal cancer tissues than in adjacent non-cancerous tissues ($P$ = 0.0019; Fig. 2C).

## Differentially expressed genes associated with HERV6196 are significantly enriched in multiple pathways related to CRC

Gene Ontology (GO) and Kyoto Encyclopedia of Genes and Genomes (KEGG) pathway enrichment analyses were performed using the differentially expressed genes that were correlated with HERV6196. GO pathway enrichment analysis revealed that HERV6196 was associated primarily with the modulation of chemical synaptic transmission, the regulation of membrane potential, cytokine activity, and extracellular matrix structural constituents (Fig. 2D). Moreover, KEGG analysis indicated that HERV6196 was enriched primarily in the cAMP signaling pathway, the cell cycle, the IL−17 signaling pathway, and the extracellular matrix (ECM)−receptor interaction (Fig. 2E). These pathway enrichment results suggested that HERV6196 may be closely associated with the development of cancer.

## Silencing HERV6196 inhibits rectal cancer cell proliferation but promotes rectal cancer cell apoptosis

Cell functional assays were conducted to further investigate the role of HERV6196 in the occurrence and progression of rectal cancer. To evaluate the effect of HERV6196 on cell proliferation, we employed short hairpin RNA (shRNA) to inhibit HERV6196 expression (Fig. 3A). In the Cell Counting Kit-8 (CCK-8) assay, cell viability was assessed at 24 h, 48 h, 72 h, 96 h, 120 h, and 144 h. Compared with the control group, the transfected group had significantly lower proliferation at 72 h, 96 h, 120 h, and 144 h (Fig. 3B). Additionally, cell cycle and apoptosis analyses were conducted. Flow cytometry was used to assess the impact of HERV6196 knockdown on the cell cycle within 24 h, which revealed that HERV6196 knockdown significantly altered HCT116 cell cycle progression. Compared with that of sh-NC cells, the proportion of sh-HERV6196 cells in the G1 phase was increased (sh-HERV6196 vs sh-NC: 60.0% vs. 38.0%), whereas the proportion of sh-HERV6196 cells in the S phase was decreased (sh-HERV6196 vs sh-NC: 16.4% vs 31.6%). For the G2/M phase comparison, the proportion of sh-HERV6196 cells was decreased compared with sh-NC cells (sh-HERV6196 vs sh-NC: 17.0% vs 24.7%, respectively), and the proportion of sh-HERV6196 cells in the G2 + S phase was decreased compared with that of sh-NC cells (sh-HERV6196 vs sh-NC: 33.4% vs 56.3%, respectively). Compared with the control group, the knockdown group exhibited reduced S phase progression and diminished DNA synthesis (Fig. 3C). Flow cytometry was also used to detect apoptosis 48 h after transfection. In Fig. 3D, quadrant Q1 represented necrotic cells, quadrant Q2 represented late apoptotic cells and dead cells, quadrant Q3 represented early apoptotic cells, and quadrant Q4 represented viable cells. Compared with the control group, the knockdown group presented significantly higher apoptosis rates (Q2 + Q3; sh-HERV6196 vs sh-NC: 27.75% vs 14.08%; (Fig. 3D). The proportion of early and late apoptotic sh-HERV6196 cells was increased compared with sh-NC cells ($P$ = 0.0039). These results indicated that HERV6196 promoted rectal cancer cell proliferation but inhibited rectal cancer cell apoptosis.

## Silencing HERV6196 inhibits the colony formation and migration of rectal cancer cells

A colony formation assay was used to compare the colony formation efficiency of the transfected and control groups. Compared with the control group, the sh-HERV6196-transfected group presented significantly lower colony formation ability (Fig. 4A). In the control group, cell migration occurred at a faster rate, and scratch healing was more pronounced. Conversely, the knockdown group exhibited considerably slower cell migration and a delayed scratch healing process. These findings suggested that the knockdown of HERV6196 inhibits the migration ability of HCT116 cells ($P$ < 0.05; Fig. 4B). Transwell invasion assays were used to assess the invasive migration ability of

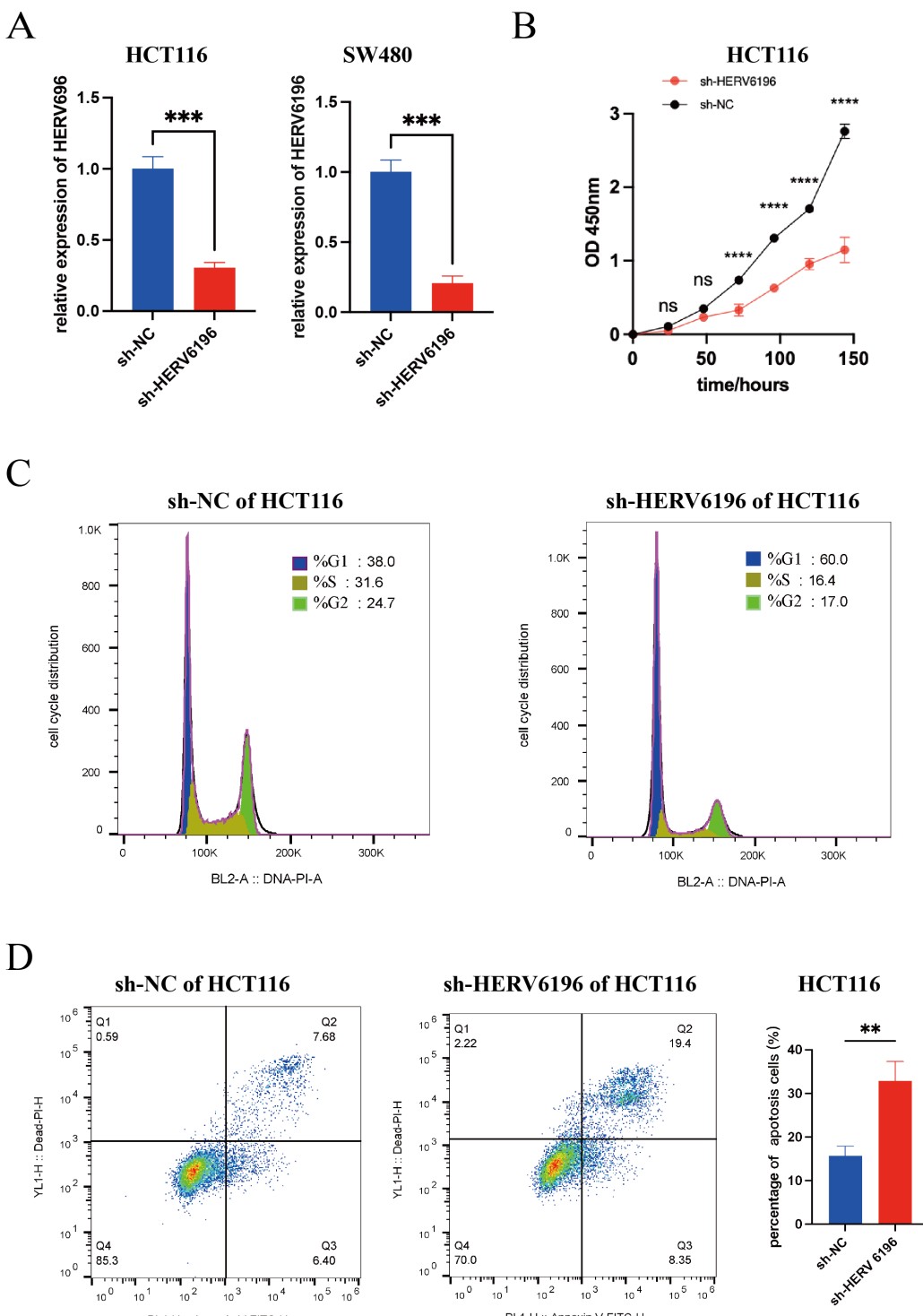

**FIG 3** Silencing HERV6196 inhibited the proliferation of HCT116 cells and promoted their apoptosis. (A) Knockdown efficiency of shRNA for interfering with the expression of HERV6196 gene transcripts in HCT116 cells and SW480 cells. (B) The CCK-8 assay was used to detect the cell proliferation status of HCT116 cells with silencing HERV6196 (red dots) compared to the normal control group (black dots). *$P < 0.05$, **$P < 0.01$, ***$P < 0.001$, and ****$P < 0.0001$; ns indicated no significant differences. (C) The cell cycle distribution was assessed in HCT116 cells, comparing the normal control group (sh-NC) with the HERV6196-deficient group (sh-HERV6196). (D) The apoptotic status of sh-NC and sh-HERV6196 in HCT116 cells was assessed, and a quantitative analysis of the percentage of apoptosis was conducted. Red represented sh-HERV6196 in HCT116 cells, and blue represented sh-NC in HCT116 cells. *$P < 0.05$, **$P < 0.01$, ***$P < 0.001$, and ****$P < 0.0001$.

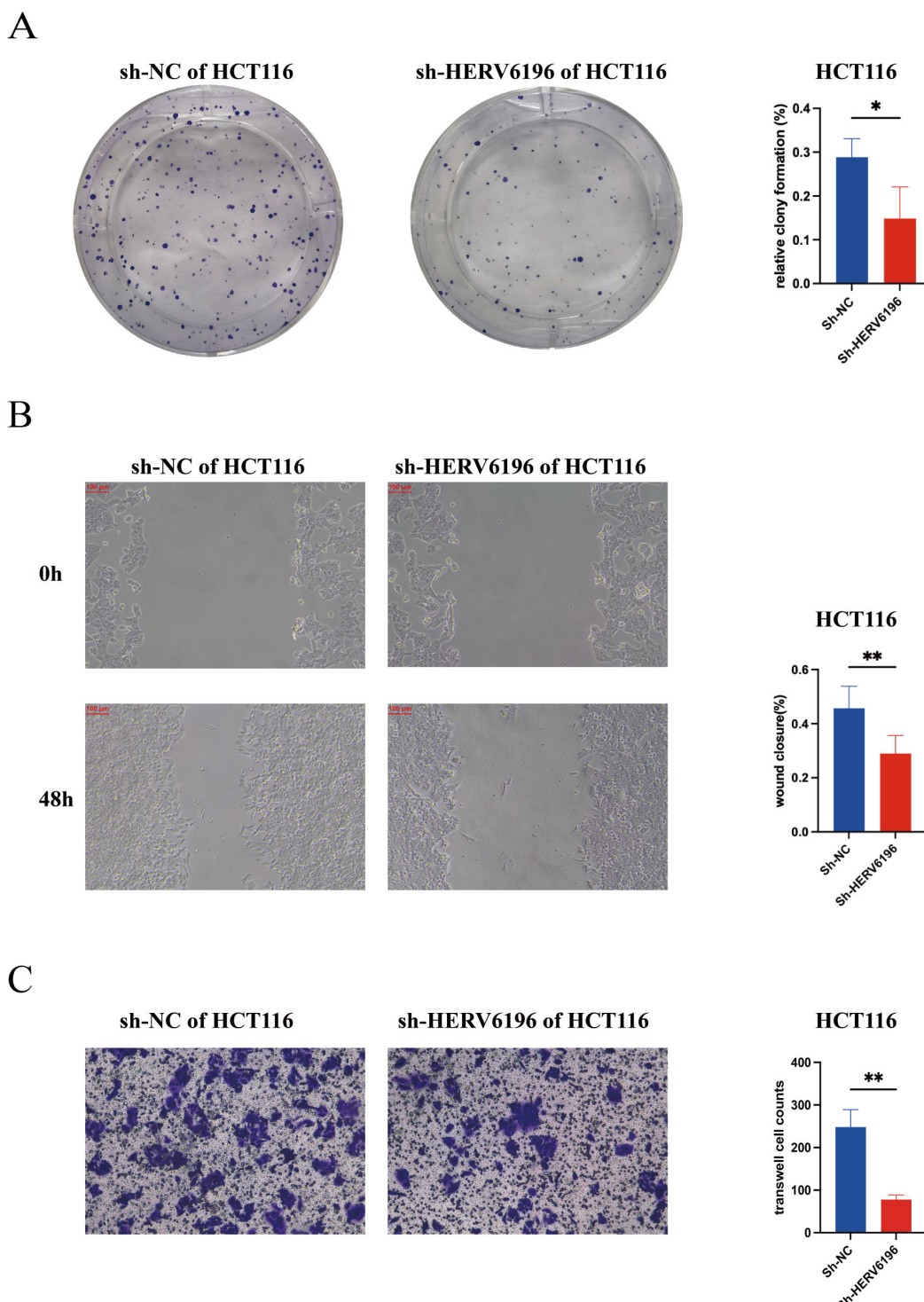

**FIG 4** Silencing HERV6196 inhibited the colony formation and migration of HCT116 cells. (A) The colony formation assay of the normal control group (sh-NC) and the HERV6196-deficient group (sh-HERV6196) in HCT116 cells was assessed, and a quantitative analysis of the percentage of colony formation was conducted. Red represented sh-HERV6196 in HCT116 cells, and blue represented sh-NC in HCT116 cells. *$P < 0.05$, **$P < 0.01$, ***$P < 0.001$, and ****$P < 0.0001$. (B) The scratch assays for sh-NC and sh-HERV6196 in HCT116 cells were performed at 0 h and 48 h, and a quantitative analysis of the percentage of wound closure was conducted. (C) The transwell invasion assay of the sh-NC and sh-HERV6196 in HCT116 cells was assessed, and a quantitative analysis of transwell cell counts was conducted.

HCT116 cells. Compared with the control cells, the HERV6196 knockdown cells exhibited decreased invasion ability (Fig. 4B). These results indicated that HERV6196 promotes the proliferation, migration, and invasion of rectal cancer cells.

## HERV6196 may act as a potential enhancer to influence the occurrence of cancer by affecting neighboring genes

Analysis of the GeneHancer database revealed that HERV6196 has two enhancer fragments, namely, GH01J221965 and GH01J221969. CRC chromatin immunoprecipitation followed by sequencing (ChIP-seq) and assay for transposase-accessible chromatin using sequencing (ATAC-seq) data were downloaded and visualized using the Integrative Genomics Viewer (IGV). Compared with adjacent normal tissues, the ChIP-seq results revealed that CRC tissues presented a greater signal intensity for HERV6196. In addition, ATAC-seq also revealed a high signal strength of HERV6196 in CRC tissues. Thus, these findings suggested that HERV6196 may function as a potential enhancer (Fig. 5A). The ChIP-seq results for the adjacent non-cancerous tissues are displayed in Fig. S1.

In order to verify the enhancer function of HERV6196, a dual luciferase reporter assay was performed. The dual-luciferase reporter assay demonstrated that the relative luciferase activity of the pGL4-HERV6196 group, which carries the HERV6196 sequence, was significantly higher than that of the null-loaded pGL4-control group ($P < 0.01$; Fig. 5B). This result confirms the enhancer activity of the HERV6196 sequence and suggests its potential involvement in rectal cancer progression through the regulation of neighboring genes.

To further investigate the mechanism by which HERV6196 promotes rectal cancer occurrence, we analyzed the neighboring genes of HERV6196. Correlation analysis revealed that HERV6196 was positively correlated with various neighboring genes, including *ADAMTS*, *BLACT1*, *DTL*, *KIF14*, *LEMD1*, *NEK2*, *LINC02474*, and *KIF26B* (Fig. S2). We subsequently confirmed the expression levels of neighboring genes following HERV6196 knockout in HCT116 cells. The knockdown sequence of HERV6196 (sh-HERV6196) was identified and validated via RT-qPCR. RT-qPCR analysis revealed a significant decrease in the expression of neighboring genes in the transfected group compared with the control group (Fig. 5C). Furthermore, western blot analysis revealed that the expression of neighboring genes was reduced following HERV6196 knockdown (Fig. 6A through E). These results suggested that HERV may function as an enhancer, promoting the development of rectal cancer by influencing neighboring oncogenes.

## DISCUSSION

In the past, HERVs were regarded as "junk DNA" owing to limitations in sequencing technology and bioinformatics software (18). However, several studies have demonstrated that HERVs are involved in gallbladder cancer, chronic lymphocytic leukemia, hepatocellular carcinoma, esophageal squamous cell carcinoma, and glioblastoma, highlighting significant differences and imbalances associated with cancer (19–23). With the advancement of multiomics big data, the functional roles of HERVs in various diseases, particularly cancer, have garnered increasing attention as a new area of research. To date, no comprehensive study has explored the locus-specific characteristics and mechanisms of HERVs in CRC. In the present study, we conducted a comprehensive analysis of HERVs in rectal cancer via ERVmap, a locus-specific identification pipeline that includes 3,220 HERVs, which revealed significant dysregulation of HERVs in rectal cancer, with 25 upregulated HERVs and 7 downregulated HERVs. Moreover, these 32 differential HERVs were identified across all three data sets, confirming the reliability of the differential analysis results. We further validated the results via RT-qPCR and confirmed their accuracy. Additionally, we collected clinical samples for precise quantification via ddPCR. Notably, HERV6196 was highly expressed in cancerous tissues but had low expression levels in adjacent non-cancerous tissues, a result that is consistent with previous studies. According to the ERVmap database, HERV6196 belongs to the HERVH type. Alves et al. reported that the HERVH gene on the X chromosome is selectively

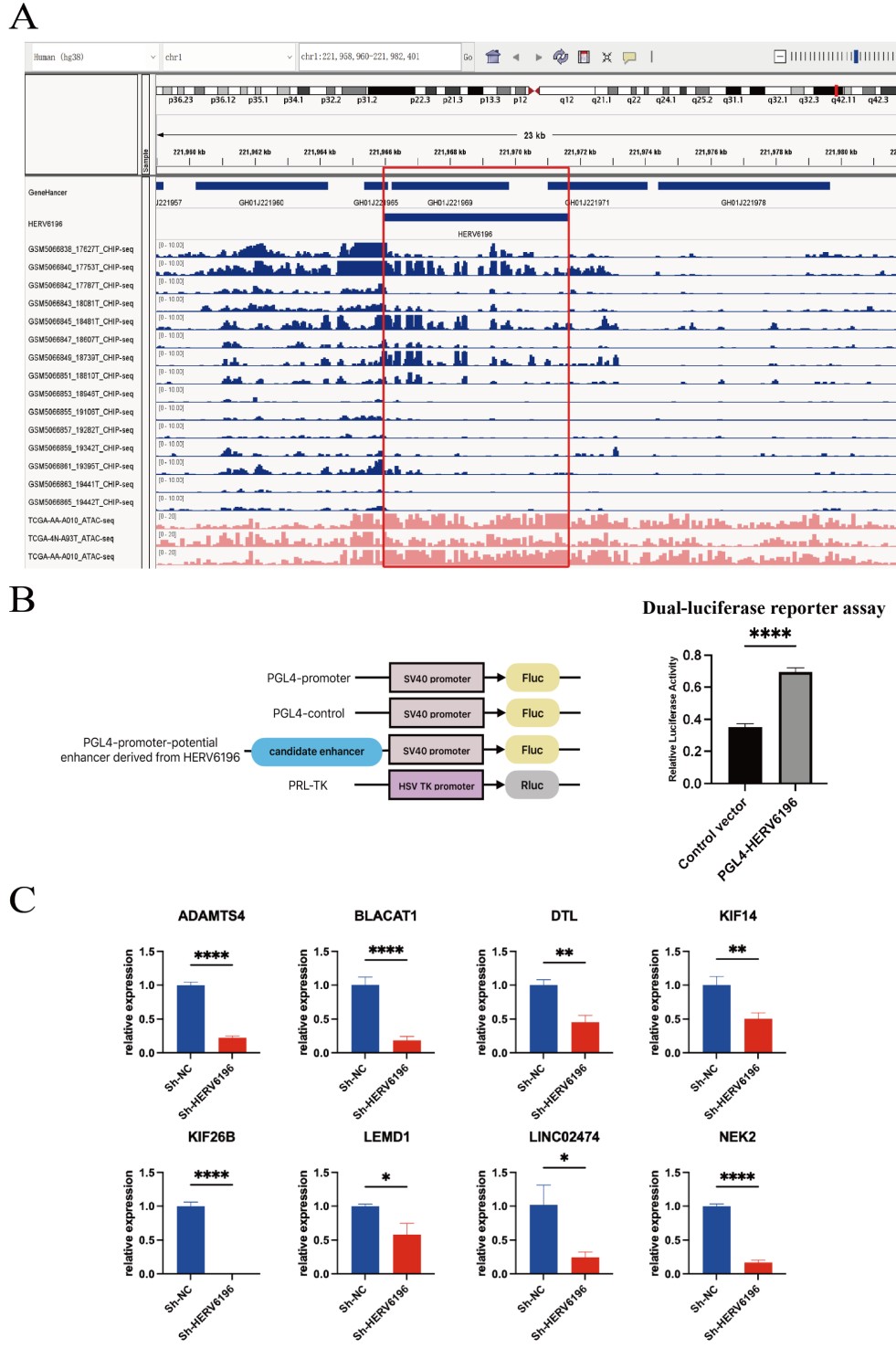

**FIG 5** HERV6196 influenced neighboring genes as a potential enhancer. (A) GeneHancer, ChIP-seq, and ATAC-seq IGV profiles of HERV6196 as a potential enhancer. The 15 ChIP-seq data sets represented CRC tissues depicted in blue, and three ATAC-seq data sets from The Cancer Genome Atlas (TCGA) depicted in red were used. The red box highlighted the ChIP-seq and ATAC-seq signal peaks in the HERV6196 region. (B) Schematic of dual-luciferase reporter assay. The PGL4 promoter vector contains an SV40 promoter upstream of the luciferase gene. The integration of herpes viruses that have been identified as containing putative enhancer elements can be achieved by means of their insertion upstream of the promoter-luc + transcription unit. PGL4-control, containing the SV40 enhancer sequence, was utilized as a positive control. (C) Following the knockdown of HERV6196, RT-qPCR values were assessed for the *ADAMTS4*, *BLACAT1*, *DTL*, *KIF14*, *KIF26B*, *LEMD1*, *LINC02474*, and *NEK2 neighboring* genes. Red represented sh-HERV6196 in HCT116 cells, and blue represented sh-NC in HCT116 cells. *$P < 0.05$, **$P < 0.01$, ***$P < 0.001$, and ****$P < 0.0001$.

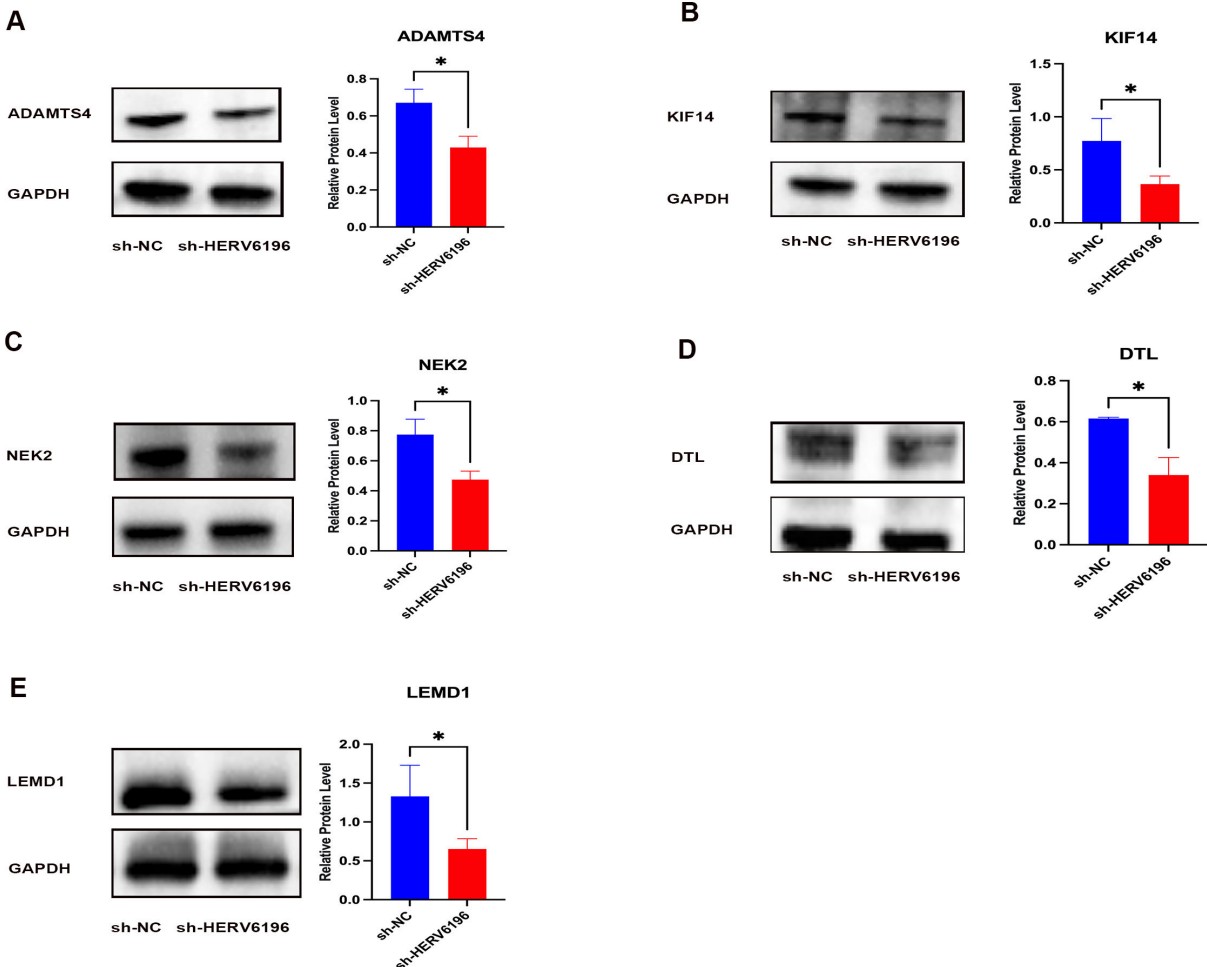

**FIG 6** Silencing HERV6196 significantly decreased the protein expression levels of neighboring genes. (A–E) Western blot analysis of the normal control group (sh-NC) and HERV6196-deficient group (sh-HERV6196) in HCT116 cells. GAPDH was used as a reference gene. The bar chart represents the grayscale values of the western blot. Red represented sh-HERV6196 in HCT116 cells, and blue represented sh-NC in HCT116 cells. *$P < 0.05$, **$P < 0.01$, ***$P < 0.001$, and ****$P < 0.0001$.

transcribed in 60% of colon cancer patients, with metastatic colon cancers presenting a greater percentage (24). Wentzensen et al. reported a greater proportion of metastatic colon cancers, and they observed the expression of HERV-H RNA sequences in various gastrointestinal tract tumors (25). These findings indicate that aberrant expression of HERVH elements is associated with cancer, a phenomenon that is particularly pronounced in cases of CRC. Thus, HERV6196 is a potential novel clinical diagnostic biomarker.

Pathway enrichment analysis of HERV6196-related differentially expressed genes revealed significant enrichment in CRC-related pathways. For example, cytokine activity and extracellular matrix structural constituent pathways affect the occurrence of CRC by regulating the tumor immune microenvironment (26, 27). The cAMP/PKA signaling pathway regulates mitochondrial autophagy, thereby promoting the onset and progression of CRC (28). In the IL-17 signaling pathway, increased expression of interleukin-17A is associated with poor prognosis of CRC patients, whereas blocking IL-17A inhibits CRC progression in preclinical cancer models (29). In addition, the ECM-receptor interaction promotes the metastatic potential of CRC (30). These results suggest a potential role for HERV in CRC. The present results demonstrated the role of HERV6196 in promoting rectal cancer cell proliferation and migration while reducing rectal cancer cell apoptosis. Thus, HERV6196 may serve as a potential biomarker and novel therapeutic target for CRC.

We also explored the mechanisms through which HERV6196 enhances the proliferation of rectal cancer. Research has demonstrated that HERVs have oncogenic effects through various mechanisms, including direct involvement in the maintenance of the tumor phenotype, inactivation of oncogenes, activation of oncogenes, mediation of cell fusion, and activation of tumor signaling pathways (31–34). HERVs have the capacity to function as promoters or enhancers, thereby triggering the activation of host genes and promoting the expression of cancer-associated genes (11). Jönsson et al. demonstrated that the LTR of HERVs acts as an alternative promoter to drive the expression of host genes (35). The activated LTR of HERV functions as an alternative promoter for neighboring genes, exerting transcriptional regulation on DNA methylation processes. Similarly, the inactivation of HERVs has been shown to directly alter gene expression in AML cell lines, suggesting that HERVs, acting as enhancers, are utilized by cancer cells to drive tumor heterogeneity and evolution (14). However, the role of HERVs in rectal cancer remains underreported in the scientific literature. In the present study, we conducted ChIP-seq and ATAC-seq analyses on HERV6196, and we utilized the GeneHancer (36) database, a comprehensive and integrated enhancer database. The present findings indicated that two segments in GeneHancer overlapped with HERV6196, suggesting that HERV6196 has potential enhancer functionality. Moreover, ChIP-seq and ATAC-seq analyses revealed significantly increased HERV6196 signals, further confirming the enhancer function of HERV6196.

Further exploration revealed that HERV6196 was positively correlated with several neighboring genes, such as *NEK2*, *LINC02474*, *LEMD1*, and ADAMTS4. Suzuki et al. (37) demonstrated that *NEK2* regulates cell division and mitosis through centrosome division. The combination of *NEK2* small interfering RNA and cisplatin has been shown to inhibit the growth of rectal cancer cells. Furthermore, elevated *NEK2* has been demonstrated to play a pivotal role in tumorigenesis and tumor progression by regulating chromosomal instability and aneuploidy, signaling pathways, mRNA selective splicing, p53, ciliolysis, and tumor immune escape (38). The *LINC02474* oncogene prevents apoptosis and promotes metastasis in CRC by inhibiting *GZMB* expression, a process that is associated with the poor prognosis of rectal cancer patients (39–42). Moreover, *LEMD1* promotes the migration of CRC cells through the RhoA/ROCK signaling pathway (43). In addition, high *KIF26B* expression is significantly associated with shorter survival of CRC patients, and the cancer-promoting roles of *KIF14* (44) in rectal cancer have been confirmed. In addition to its role in the growth of early-stage lung cancer, *ADAMTS4* promotes tumor progression in hepatocellular carcinoma, lung cancer, and CRC (45). Furthermore, we confirmed that HERV6196 acts as an enhancer, promoting the expression of oncogenes and contributing to the development of rectal cancer. These findings offer new insights into the mechanisms underlying the development of CRC.

In summary, the present study demonstrated that HERV6196 is significantly upregulated in various rectal cancer cell lines. Functional analyses further demonstrated that the knockdown of HERV6196 expression significantly inhibits tumor proliferation, invasion, and migration. These findings underscore the association between HERV6196 expression and the progression of rectal cancer, suggesting that HERV6196 may serve as a novel biomarker for this disease. Additionally, HERV6196 may function as an enhancer, playing a role in the regulatory mechanisms of CRC. We will further explore the enhancer mechanisms in future studies using methods such as reporter assays and chromosome conformation capture. Future research is needed to explore the upstream mechanisms influencing HERV occurrence. Additionally, HERV6196 should be validated as a novel diagnostic biomarker for CRC using a larger clinical sample size.

## Conclusions

HERVs are implicated in the development of various cancers. However, comprehensive single-site HERV studies in CRC are lacking. In the present study, we validated the differential expression of HERVs in rectal cancer via ERVmap, RT-qPCR, and ddPCR. Subsequent functional analysis revealed that the knockdown of HERV6196 expression

significantly inhibits tumor proliferation, invasion, and migration. These studies suggest that HERV6196 could serve as a novel biomarker for rectal cancer. Mechanistically, HERV may function as an enhancer that promotes neighboring genes, such as *NEK2*, *LINC02474*, *LEMD1*, and ADAMTS4. In conclusion, the present research highlights the potentially crucial role of human endogenous retroviruses in the biology of rectal cancer.

## MATERIALS AND METHODS

### Data collection

The raw RNA sequencing data for rectal cancer patients were extracted from the NCBI SRA database (https://www.ncbi.nlm.nih.gov/sra). The following data sets were utilized to identify differentially expressed HERVs: GSE50760, which comprises 18 cancer tissues and 18 adjacent normal tissues; GSE104836, which includes 10 cancer tissues and 10 adjacent normal tissues; and GSE142279, which contains 20 cancer tissues and 20 adjacent normal tissues.

### Sample collection

Plasma and tissues were obtained from patients with rectal cancer at Fujian Cancer Hospital between January 2021 and December 2024. All samples were obtained prior to treatment, and a total of 18 cancerous and adjacent normal tissues were collected for ddPCR. Tissue gDNA was extracted from formalin-fixed paraffin-embedded (FFPE) or fresh tumor samples via a QIAamp FFPE tissue kit (Qiagen, Hilden, Germany).

### Calculation of HERV gene transcript expression using ERVmap

The expression of HERV gene transcripts was analyzed via ERVmap software. Specifically, the RNA sequencing reads were first mapped to the human genome (hg38) via the Burrows–Wheeler Aligner. The aligned reads were subsequently filtered according to the stringent criteria of ERVmap via a script designed specifically for ERVmap, and the reads were subsequently mapped to ERV loci. Finally, the expression of HERV gene transcripts was normalized according to the ERVmap protocol.

### Differential expression of HERV gene transcripts

Differential expression analysis of HERVs in rectal cancer tissues vs adjacent normal tissues was conducted via the DESeq2 R package. The data sets analyzed included the GSE50760, GSE104836, and GSE142279 data sets. The screening criteria were set at |log2FoldChange| > 1 and padj < 0.05. To visualize expression patterns, TBtools software was utilized (https://github.com/CJ-Chen/TBtools) to generate a heatmap of HERV gene transcript expression levels.

### Correlation analysis of HERVs with neighboring genes

To investigate the relationship between HERVs and neighboring genes, the expression of genes and HERVs was initially computed via ERVmap software. Finally, Spearman correlation analysis was performed to assess the relationship between HERVs and adjacent genes, applying thresholds of a correlation coefficient |R| > 0.3 and *P* < 0.05.

### GO and KEGG pathway enrichment analyses

The clusterProfiler package in R was utilized to conduct GO and KEGG enrichment analyses. Additionally, the results were visualized via histograms.

## Cell lines and culture

Human rectal adenocarcinoma cell lines (SW480, HCT116, HT29, LOVO, and SW620) and normal human colorectal epithelial cells (HCoEpiC) were purchased from Wuhan Pricella Biotechnology Co., Ltd., China. All the cells were cultured in Dulbecco's modified Eagle's medium (DMEM; Thermo Fisher Scientific) supplemented with 10% fetal bovine serum (FBS; Meisen Chinese Tissue Culture Collections) at 37°C in a 5% $CO_2$ atmosphere.

## RNA extraction and reverse transcription

RNA extraction was performed via the TRIzol Up Plus RNA Kit (Beyotime) according to the manufacturer's instructions. The purity and concentration of the extracted RNA were determined via UV spectrophotometry, with an OD260/OD280 ratio (R value) within the range of 1.8–2.2, indicating high purity. The RNA was subsequently reverse transcribed to cDNA via the HiScript II Q RT SuperMix for qPCR (+gDNA wiper) kit (Novozymes).

## RT-qPCR and primer design

RT-qPCR was conducted using the Hieff qPCR SYBR Green Master Mix Kit (Yeasen) and an ABI 7500 Fast Real-Time PCR system (Applied Biosystems). The cycling conditions were as follows: initial denaturation at 95°C for 10 min, followed by 40 cycles of 95°C for 15 s and 60°C for 35 s. The relative expression of RNA was calculated via the $2^{-\Delta\Delta Ct}$ method. Each experiment was independently repeated three times. The cDNA products were stored at −80°C. Primers for HERVs and genes were designed via the NCBI Primer-BLAST tool (https://www.ncbi.nlm.nih.gov/tools/primer-blast/). Table S2 lists the primers used in this study.

## Droplet digital PCR

DdPCR was performed via the Probe One-Step RT−ddPCR Advanced Kit (Bio-Rad, Hercules, CA, USA). The reaction mixture had a final volume of 22 µL, with 5.5 µL of super mixture, 2.2 µL of 20 U/µL reverse transcriptase, 1.1 µL of 15 mM dithiothreitol, 900 nM Oropouche virus (OROV) primer, 250 nM probe, 2 µL of 20× RPP30 analysis, and 7 µL of RNA template. A QX200 droplet generator (Bio-Rad, Hercules, CA, USA) was used to convert 20 µL of each reaction mixture into droplets. The droplet aliquots were transferred to a 96-well plate, sealed, and processed in a C1000 touch thermal cycler (Bio-Rad). The following cycling protocol was used: holding at 25°C for 3 min; reverse transcription at 50°C for 60 min; enzyme activation at 95°C for 10 min; 50 cycles of denaturation at 95°C for 30 s and annealing/extension at 60°C for 60 s; enzyme inactivation at 98°C for 10 min; and holding at 4°C for 30 min. The amplified samples were then transferred and read in the FAM (OROV) and HEX (RPP30) channels via a QX200 reader (Bio-Rad, Hercules, CA, USA). The data were analyzed via QXManager 1.2 standard edition software (Bio-Rad, Hercules, CA, USA) and expressed as copy number/µL (cp/µL) in the ddPCR.

## CCK-8 assay

The cells were seeded into 96-well plates and incubated for 24 h, 48 h, 72 h, 96 h, 120 h, or 144 h. After treatment, 10 µL of CCK-8 reagent (Beyotime Biotechnology) was added to each well and incubated at 37°C for 1−4 h. The absorbance was measured at 450 nm via a microplate reader to assess cell viability.

## Colony formation assay

The cells were plated in six-well plates at a low density (500 cells/well) and cultured for 10−14 days. Colonies were fixed with 4% paraformaldehyde, stained with crystal violet, and counted to evaluate their proliferative capacity.

## Invasion assay

Transwell chambers coated with Matrigel were used for the invasion assay. The cells suspended in serum-free medium were added to the upper chamber, and medium containing 10% FBS was added to the lower chamber. After incubation for 48 h, the invading cells were fixed, stained, and counted under a microscope.

## Scratch assay

A monolayer of cells was scratched with a pipette tip to create a wound. The cells were cultured in medium with reduced serum, and images were captured at 0 h and 48 h. The migration distance was measured to assess cell motility.

## Apoptosis assay

The cells were stained with Annexin V-FITC and propidium iodide (Cell Cycle and Apoptosis Detection Kit, Beyotime) according to the manufacturer's protocol. Flow cytometry was used to quantify apoptotic cells by measuring fluorescence signals.

## Identification of enhancers

ChIP-seq data for three CRC and adjacent tissue pairs were downloaded from the GSE166254 database. We downloaded the hg19 normalized bw files from GSE166254 and used CrossMap to convert the hg19 normalized bw files to hg38 normalized bw files, which were then exported to IGV for display. Additionally, ATAC-seq data for three CRC cases were obtained from The Cancer Genome Atlas (TCGA). Both the ChIP-seq and ATAC-seq data were visualized via the IGV. Potential enhancer elements were also downloaded from the GeneHancer database.

## Western blot analysis

Protein lysates were extracted from cells via radioimmunoprecipitation assay (RIPA) lysis buffer supplemented with protease and phosphatase inhibitors. Protein concentrations were quantified via the bicinchoninic acid assay. Equal amounts of protein (20−40 μg) were resolved on SDS−PAGE gels and transferred onto polyvinylidene difluoride membranes. The membranes were blocked with 5% nonfat milk or BSA in TBST buffer and incubated overnight at 4°C with primary antibodies. After washing, the membranes were incubated with horseradish peroxidase-conjugated secondary antibodies, and signals were detected via enhanced chemiluminescence. Band intensities were quantified by ImageJ software for further analysis.

## Dual-luciferase reporter assay

The detection of HERV6196 enhancer activity was accomplished through the utilization of a dual luciferase reporter gene assay system. Initially, the HERV6196 sequence was engineered into the pGL4 luciferase reporter vector, thereby constructing the recombinant reporter plasmid pGL4-HERV6196. Subsequently, the aforementioned vectors and the empty vectors were transferred into 24-well plates of 293T cells using Lipofectamine 3000 (Thermo Fisher; four wells for each vector). Four replicate wells were utilized for each vector. Concurrently, the pRL-TK Luciferase Vector was utilized as an internal reference and co-transfected with the aforementioned luciferase reporter vectors. Following a 24 h transfection period, the activities of Firefly Luciferase and Renilla Luciferase were measured sequentially in accordance with the Promega instructions. This experiment was repeated on three occasions, and the ratio of Firefly Luciferase activity to Renilla Luciferase activity was employed for normalization.

## Statistical analysis

Statistical analysis was conducted via GraphPad Prism (version 8.0.2) and R (version 4.2.1). The data were analyzed via Student's $t$-test for comparisons between two groups. Each experiment was independently repeated three times. The Wilcoxon signed-rank test was employed to compare differential expression between cancerous and normal samples. Correlation analysis was performed via Spearman's rank correlation coefficient, and a $P$-value less than 0.05 was considered statistically significant.

## ACKNOWLEDGMENTS

We acknowledge the NCBI Sequence Read Archive (SRA) database for providing platforms and contributors for uploading meaningful data sets. We acknowledge the provision of biological samples by Tumor Biobank of Fujian Cancer Hospital. We thank Dr. Yu Xiao for his valuable technical support during the revision of this manuscript.

This work was supported by the Fujian Province Gastrointestinal, Respiratory, and Genitourinary Malignant Tumor Radiotherapy Radiation and Treatment Clinical Medical Research Center (2021Y2014), the Fujian Provincial Clinical Medical Research Center for Tumor Precision Radiotherapy (2020Y20101), the Fujian Province Science and Technology Innovation Joint Funding Project (2021Y9216), the Fujian Province Natural Science Foundation (2021J01438 and 2022J01433), Fujian Cancer Hospital In-Hospital Funding Program (2022YNG06 and 2023YNPT00), the Fujian Provincial Clinical Research Center for Cancer Radiotherapy and Immunotherapy (2020Y2012), the National Clinical Key Specialty Construction Program, and Sichuan Medical Association youth innovation project, Q2024048.

Y.-X.G.: investigation, formal analysis, data curation, writing—original draft, and writing—review and editing. X.J.: conceptualization, data curation, software, investigation, and supervision. Z.-Y.W.: investigation and validation. Y.-L.Y.: Data curation. L.-D.S.: data curation. J.-M.W.: Conceptualization, data curation, funding acquisition, investigation, and writing—review and editing. J.-X.W.: conceptualization, data curation, funding acquisition, investigation, project administration, resources, supervision, writing—review and editing.

## AUTHOR AFFILIATIONS

[1]Department of Radiation Oncology, Clinical Oncology School of Fujian Medical University, Fujian Cancer Hospital, Fuzhou, People's Republic of China
[2]Department of Tumor Radiotherapy, Hezhou People's Hospital, Hezhou, People's Republic of China
[3]Innovation Center for Cancer Research, Clinical Oncology School of Fujian Medical University, Fujian Cancer Hospital, Fuzhou, People's Republic of China
[4]Fujian Key Laboratory of Advanced Technology for Cancer Screening and Early Diagnosis, Fuzhou, People's Republic of China

## AUTHOR ORCIDs

Yi-Xiu Gan http://orcid.org/0000-0002-4332-1934
Jian-Min Wang http://orcid.org/0000-0002-7395-3270
Jun-Xin Wu http://orcid.org/0000-0003-1047-2338

## FUNDING

| Funder | Grant(s) | Author(s) |
| --- | --- | --- |
| Fujian Province Gastrointestinal, Respiratory, and Genitourinary Malignant Tumor Radiotherapy Radiation and Treatment Clinical Medical Research Center | 2021Y2014 | Jun-Xin Wu |
| Fujian Provincial Clinical Medical Research Center | 2020Y20101 | Jun-Xin Wu |

| Funder | Grant(s) | Author(s) |
| --- | --- | --- |
| Fujian Province Science and Technology Innovation Joint Funding Project | 2021Y9216 | Jun-Xin Wu |
| Fujian Province Natural Science Foundation | 2021J01438, 2022J01433 | Jun-Xin Wu |
| Fujian Cancer Hospital In-Hospital Funding Program | 2022YNG06, 2023YNPT00 | Jun-Xin Wu |
| Fujian Provincial Clinical Research Center for Cancer Radiotherapy and Immunotherapy | 2020Y2012 | Xin Jiang |
| National Clinical Key Specialty Construction Program, and Sichuan Medical Association youth innovation project | Q2024048 | Jun-Xin Wu Xin Jiang |

## AUTHOR CONTRIBUTIONS

Yi-Xiu Gan, Data curation, Formal analysis, Investigation, Writing – original draft | Xin Jiang, Conceptualization, Data curation, Investigation, Software, Supervision | Zhi-Yu Wang, Investigation, Validation | Yi-Lin Yu, Data curation | Ling-Dong Shao, Data curation | Jian-Min Wang, Conceptualization, Data curation, Funding acquisition, Investigation, Writing – review and editing | Jun-Xin Wu, Conceptualization, Data curation, Funding acquisition, Investigation, Resources, Supervision, Writing – review and editing

## DATA AVAILABILITY

The processed numerical data generated in this study have been deposited in the Figshare repository under the permanent DOI 10.6084/m9.figshare.30390199.

## ETHICS APPROVAL

This study was implemented in accordance with the principle of the Declaration of Helsinki and approved by the Fujian Cancer Hospital (approval no. K2023-181-01), and written informed consent was obtained from all patients before enrolling in the research program.

## ADDITIONAL FILES

The following material is available online.

### Supplemental Material

**Supplemental material (Spectrum00788-25-s0001.pdf).** Fig. S1 and S2; Table S2; Data S1.
**Table S1 (Spectrum00788-25-s0002.xlsx).** The differentially expressed HERVs using the DESeq2 package.

### Open Peer Review

**PEER REVIEW HISTORY (review-history.pdf).** An accounting of the reviewer comments and feedback.

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
