## [Reviewer comments · Microbiology Spectrum]

Microbiology Spectrum

HERV6196 as an enhancer with oncogenic potential in rectal cancer

Yi-Xiu Gan, Xin Jiang, Zhi-Yu Wang, Yi-Lin Yu, Ling-Dong Shao, Jianmin Wang, and Jun-Xin Wu

Corresponding Author(s): Jun-Xin Wu, Department of Radiation Oncology, Clinical oncology School of Fujian Medical University, Fujian Cancer Hospital, Fuzhou, 350014, P.R. China.

Review Timeline:

Submission Date:	March 16, 2025
Editorial Decision:	April 25, 2025
Revision Received:	August 9, 2025
Editorial Decision:	October 14, 2025
Revision Received:	October 25, 2025
Accepted:	November 8, 2025

Editor: Hyun Jin Kwun

Reviewer(s): Disclosure of reviewer identity is with reference to reviewer comments included in decision letter(s). The following individuals involved in review of your submission have agreed to reveal their identity: Michal Izydorczyk (Reviewer #1)

Transaction Report:

DOI: <https://doi.org/10.1128/spectrum.00788-25>

Re: Spectrum00788-25 (**HERV6196 as an enhancer with oncogenic potential in rectal cancer**)

Dear Dr. Jun-Xin Wu:

Thank you for the privilege of reviewing your work. Below you will find instructions from the Spectrum editorial office, and the reviewer comments.

Revision Guidelines

Sincerely,
Hyun Jin Kwun
Editor
Microbiology Spectrum

Reviewer #1 (Comments for the Author):

The study by Gan et al. provides a comprehensive and compelling exploration of the role of human endogenous retroviruses (HERVs), particularly HERV6196, in the development and progression of rectal cancer. By integrating bioinformatics analyses with experimental validation, the authors present a well-rounded investigation that significantly advances our understanding of how HERVs contribute to oncogenesis. The study is particularly notable for its innovative focus on locus-specific profiling of HERVs, overcoming the challenges posed by their repetitive nature, and for its potential clinical implications in colorectal cancer (CRC) diagnosis and therapy.

One of the key strengths of this work is its methodological rigor. The authors employed ERVmap to analyze RNA-seq data from multiple GEO datasets, identifying differentially expressed HERVs in rectal cancer tissues compared to adjacent normal tissues. This bioinformatics approach was complemented by qRT-PCR and ddPCR validation, ensuring the reliability of the findings. The functional assays, including proliferation, apoptosis, and migration experiments, further solidified the role of HERV6196 in promoting cancer progression. The integration of ChIP-seq and ATAC-seq data to demonstrate HERV6196's enhancer activity and its influence on neighboring oncogenes (e.g., NEK2, LINC02474) was particularly insightful, offering a mechanistic explanation for its oncogenic effects.

The study's clinical relevance is another major highlight. The identification of HERV6196 as a potential biomarker for rectal cancer addresses a critical need for improved diagnostic tools, given the limitations of current screening methods. The authors' findings suggest that HERV6196 could serve as a novel target for therapeutic intervention, opening new avenues for precision medicine in CRC.

However, while the knockdown experiments show HERV6196's role in cancer phenotypes, the exact direct mechanistic link (e.g., whether HERV6196's enhancer activity alone drives oncogenesis) could be further probed (e.g., via reporter assays or chromatin conformation studies). Further, the ddPCR validation in 18 patient samples is promising but preliminary. A larger cohort or survival analysis would solidify its biomarker potential.

While the study is robust, there are minor areas where further exploration could enhance its impact. For instance, the clinical validation cohort of 18 paired samples, though sufficient for initial findings, would benefit from expansion to strengthen the translational potential of HERV6196 as a biomarker. Additionally, investigating the upstream regulatory mechanisms driving HERV6196 dysregulation—such as epigenetic modifications or environmental factors—could provide deeper insights into its role in cancer initiation and progression.

As for language, it is technically sound but would benefit from professional editing to polish clarity and flow.

In conclusion, this study makes a significant contribution to the field by elucidating the oncogenic potential of HERV6196 in rectal cancer. The combination of bioinformatics, experimental validation, and mechanistic exploration provides a strong foundation for the authors' conclusions. The work is well-written, methodologically sound, and clinically relevant, making it a valuable addition to the literature on CRC and retrovirology.

Reviewer #2 (Comments for the Author):

The manuscript aims to identify HERV transcripts in a colorectal cancer dataset deposited in GEO using bioinformatics tools such as ERVmap and other programs to characterize the potential role of a specific HERV sequence or location in cancer cell lines.

- This manuscript is poorly written and needs extensive revision in scientific writing and figure quality. I suggest that authors utilize professional services.
- Verify that the references are properly cited in the main text.
- Expression of HERV gene transcripts' or 'Transcripts' would be better to use throughout the manuscript instead of using 'HERV expression'.
- What is HERV6169 and what does it mean? The authors should explain the nomenclature they used for general readers. Can we assume that this is HERV-W because it is located on chromosome 1? Can authors at least briefly discuss the HERV type they are analyzing?
- All figures (from 1 to 10) should be correctly labeled, so readers can understand quickly without having to check the figure legend each time. All figures' legends should be elaborated. Some of the figure legends do not specify which cells they are analyzing.
- Fig. 1 D-F is not readable. Fig.1B does not show HERV6196 as upregulated, and is there any reason to choose HERV6196 loci for these analyses? HERV5290 is highly upregulated in all CRC tissues. What is the reason that it was not chosen for further study? What is the interpretation for the proviral HERV that was downregulated in a specific locus?
- Fig. 6: please label them appropriately. Fig6A used siRNA, while 6B-G used shRNA according to the figure legend? Please label 6C and 6D and analyze cell cycles and how much S phase changes occur. What are the meanings of these colors? The y-axis (count) is different. Please make it the same scale. If the y-axis is adjusted to the same scale, will the knockdown condition

still decrease the S phase? Can authors analyze dead cells using this flow data by gating them again? To simplify the interpretation of these numbers, label 6E and 6F on the figures. Are these early and late apoptosis seen in 6G?

- Please label Fig. 7 within the Figures (like labeling parts of an image). What are these cells and which cell lines are used? Fig. 7i-7k lacks convincing evidence. Is this an invasion or migration assay? It is a different scientific term. Matrigel was used in the method, but the figure legend indicates migration.

- Fig. 10 is not convincing. The ECL method for western is not quantitative. Is there any statistical analysis? What makes this significant?

- Material methods: Please specify sources for all reagents. What is the original source of cells used for this study? What does it mean if this comes from laboratory stocks?

- The authors downloaded ChIP-seq data from GSE166254. Why were only three samples of tumors and adjacent tissues analyzed? There are more samples and they should be included. Are these ChIP data from samples with upregulated HERV6169? To make it easier for readers to understand, please give an explanation of the ChIP condition and the reason why you used these data. This is the data from ChIP-seq with histone H3 acetylation of Lysine 27, which is a marker for active gene regulation. And please explain briefly why you used ChIP and ATAC-seq data and compared.

- Too many figures. Figs 1-2, Fig.3-5, and Fig. 8-10 can be combined.

Responses to the comments of the Reviewers

We sincerely thank the editor and all the reviewers for their valuable feedback, which we utilized to improve the quality of the manuscript. The reviewers' comments are displayed in italics below, with specific issues numbered, and our responses provided in blue text.

Response to Reviewer #1

Reviewer #1:

1. The study by Gan et al. provides a comprehensive and compelling exploration of the role of human endogenous retroviruses (HERVs), particularly HERV6196, in the development and progression of rectal cancer. By integrating bioinformatics analyses with experimental validation, the authors present a well-rounded investigation that significantly advances our understanding of how HERVs contribute to oncogenesis. The study is particularly notable for its innovative focus on locus-specific profiling of HERVs, overcoming the challenges posed by their repetitive nature, and for its potential clinical implications in colorectal cancer (CRC) diagnosis and therapy.

One of the key strengths of this work is its methodological rigor. The authors employed ERVmap to analyze RNA-seq data from multiple GEO datasets, identifying differentially expressed HERVs in rectal cancer tissues compared to adjacent normal tissues. This bioinformatics approach was complemented by qRT-PCR and ddPCR validation, ensuring the reliability of the findings. The functional assays, including proliferation, apoptosis, and invasion experiments, further solidified the role of HERV6196 in promoting cancer progression. The integration of ChIP-seq and ATAC-seq data to demonstrate HERV6196's enhancer activity and its influence on neighboring oncogenes (e.g., NEK2, LINC02474) was particularly insightful, offering a mechanistic explanation for its oncogenic effects.

The study's clinical relevance is another major highlight. The identification of HERV6196 as a potential biomarker for rectal cancer addresses a critical need for

improved diagnostic tools, given the limitations of current screening methods. The authors' findings suggest that HERV6196 could serve as a novel target for therapeutic intervention, opening new avenues for precision medicine in CRC.

Response:

We would like to thank you for your professional review work and encouraging comment.

2.However, while the knockdown experiments show HERV6196's role in cancer phenotypes, the exact direct mechanistic link (e.g., whether HERV6196's enhancer activity alone drives oncogenesis) could be further probed (e.g., via reporter assays or chromatin conformation studies). Further, the ddPCR validation in 18 patient samples is promising but preliminary. A larger cohort or survival analysis would solidify its biomarker potential.

Response:

Through the comprehensive analysis of the Genehancer enhancer database combined with ChIP-seq and ATAC-seq, we identified that HERV6196 has potential enhancer activity [page 14, lines 226-233]. Additionally, after knocking out HERV6196, we found that the expression of nearby oncogenes associated with CRC was also significantly reduced, indicating that HERV6196 may be a potential enhancer that promotes the occurrence of CRC [page 16, lines 256-265]. In accordance with your professional opinion, a dual luciferase assay was conducted on the function of HERV6196 enhancer, with the results indicating that enhancer activity of HERV6196. This result is presented in the results section of the manuscript [page 16, lines 250-255]. In the future, we plan to incorporate more clinical samples to further demonstrate its potential as a clinical biomarker.

3.While the study is robust, there are minor areas where further exploration could enhance its impact. For instance, the clinical validation cohort of 18 paired samples, though sufficient for initial findings, would benefit from expansion to strengthen the translational potential of HERV6196 as a biomarker. Additionally, investigating the upstream regulatory mechanisms driving HERV6196 dysregulation-such as epigenetic modifications or environmental factors-could provide deeper insights into its role in

cancer initiation and progression.

Response:

Thank you for your suggestion. In the future, we will include a larger clinical cohort to explore the translational potential of HERV6196 as a biomarker. We have added this section to the discussion part of the manuscript. In addition, we will investigate the upstream regulatory mechanisms driving the dysregulation of HERV6196 in the future. We have added this section to the discussion part of the manuscript [page 21, lines 359-368].

4.As for language, it is technically sound but would benefit from professional editing to polish clarity and flow.

Response:

According to your suggestion, we have carried out professional editing to enhance the fluency and professionalism of the manuscript.

5.In conclusion, this study makes a significant contribution to the field by elucidating the oncogenic potential of HERV6196 in rectal cancer. The combination of bioinformatics, experimental validation, and mechanistic exploration provides a strong foundation for the authors' conclusions. The work is well-written, methodologically sound, and clinically relevant, making it a valuable addition to the literature on CRC and retrovirology.

Response:

Thank you for your professional comments. In the future, we will continue to expand the clinical sample size to further investigate the translational potential of HERV6196 as a clinical marker, and to explore the upstream and downstream mechanisms by which HERV6196 promotes the occurrence of CRC. In summary, our research provides a foundation for the oncogenic potential of HERV in cancer.

Response to Reviewer #2

Reviewer #2:

1.This manuscript is poorly written and needs extensive revision in scientific writing and figure quality. I suggest that authors utilize professional services.

Response:

According to your suggestion, we have carried out professional editing to enhance the fluency and professionalism of the manuscript. In addition, we have improved the clarity and quality of all the figures in the manuscript. Thank you for your professional advice.

2. Verify that the references are properly cited in the main text.

Response:

Following your suggestion, we have revised some of the references to ensure that all content in the main text is supported by the cited literature, and we have confirmed that the format of the references is correct.

3. Expression of HERV gene transcripts' or 'Transcripts' would be better to use throughout the manuscript instead of using 'HERV expression'.

Response:

Based on your suggestion, we used 'expression of HERV gene transcripts' instead of 'HERV expression'.

4. What is HERV6169 and what does it mean? The authors should explain the nomenclature they used for general readers. Can we assume that this is HERV-W because it is located on chromosome 1? Can authors at least briefly discuss the HERV type they are analyzing?

Response:

The naming of HERV6196 comes from the ERVmap database, which defines 3,220 HERVs, one of which is HERV6196. According to the ERVmap database, HERV6196 belongs to the HERVH type. We have added 'HERV6196, a type of HERVH', to the abstract of the manuscript [page 2, line 28-30]. And we have added a discussion on the role of HERVH in colorectal cancer in the discussion section [page 18, lines 301-307].

5. All figures (from 1 to 10) should be correctly labeled, so readers can understand quickly without having to check the figure legend each time. All figures' legends should be elaborated. Some of the figure legends do not specify which cells they are analyzing.

Response:

Thank you for your professional comments. We provided detailed explanations for all the figure legends so that readers can quickly understand them without needing to check the figure legends each time. Additionally, we corrected the figure legends to specify the cells being analyzed.

6.Fig. 1 D-F is not readable. Fig.1B does not show HERV6196 as upregulated, and is there any reason to choose HERV6196 loci for these analyses? HERV5290 is highly upregulated in all CRC tissues. What is the reason that it was not chosen for further study? What is the interpretation for the proviral HERV that was downregulated in a specific locus?

Response:

We improved the clarity of Figures 1D-F to make them readable. Additionally, we also enhanced the clarity of all the images in the manuscript.

Figure 1B only shows the top five upregulated HERVs in the GSE104836. GSE104836 includes 86 upregulated HERVs, with HERV6196 ranking 11th ($\log_2\text{FoldChange}=3.34$, $\text{padj}<0.05$). We have updated Figures 1A-C to display the top 15 HERVs with the greatest upregulation and downregulation. Additionally, Figures 1A and 1C both show HERV6196 among the top five upregulated, so we believe there is justification for selecting the HERV6196 locus for these analyses.

Although HERV5290 is highly upregulated in all CRC tissues, it is located 87,374 bp away from the nearest gene LOC101928201, which lacks biological significance. This contradicts our scientific hypothesis that HERVs promote CRC occurrence by influencing neighboring genes. Therefore, we did not select HERV5290 for further research.

This study primarily focuses on upregulated HERVs that promote the occurrence of CRC by affecting adjacent oncogenes. Therefore, we have temporarily overlooked downregulated HERVs in this study; however, we believe that downregulated HERVs may serve as diagnostic markers for CRC, and we will continue to investigate this in the future.

7.Fig. 6: please label them appropriately. Fig6A used siRNA, while 6B-G used shRNA

according to the figure legend? Please label 6C and 6D and analyze cell cycles and how much S phase changes occur. What are the meanings of these colors? The y-axis (count) is different. Please make it the same scale. If the y-axis is adjusted to the same scale, will the knockdown condition still decrease the S phase? Can authors analyze dead cells using this flow data by gating them again? To simplify the interpretation of these numbers, label 6E and 6F on the figures. Are these early and late apoptosis seen in 6G?

Response:

Thank you for your careful review. We have renamed the labels correctly in the manuscript. The figures have been rearranged and are Fig. 3 now in the manuscript. Fig.3A-D all utilized shRNA.

We labeled Fig.3C. In cell cycle analysis, HERV6196 knockdown treatment significantly altered HCT116 cell cycle progression. Compared with that of sh-NC cells, the proportion of sh-HERV6196 cells in the G1 phase was increased (sh-HERV6196 vs. sh-NC: 60.0% vs. 38.0%), whereas the proportion of sh-HERV6196 cells in the S phase was decreased (sh-HERV6196 vs. sh-NC: 16.4% vs. 31.6%). For the G2/M phase comparison, the proportion of sh-HERV6196 cells was decreased compared with sh-NC cells (sh-HERV6196 vs. sh-NC: 17.0% vs. 24.7%, respectively), and the proportion of sh-HERV6196 cells in the G2+S phase was decreased compared with that of sh-NC cells (sh-HERV6196 vs. sh-NC: 33.4% vs. 56.3%, respectively). Compared with the control group, the knockdown group exhibited reduced S phase progression and diminished DNA synthesis (Fig. 3C) [page 10, lines 173-180]. This suggests that knockdown of HERV6196 inhibits the growth of tumor cells, which has been further analyzed in detail and is reflected in the manuscript. We have made the y-axis the same scale.

In Figure 3D, quadrant Q1 represents necrotic cells, quadrant Q2 represents late apoptotic cells and dead cells, quadrant Q3 represents early apoptotic cells, and quadrant Q4 represents viable Cells. Compared with the control group, the knockdown group presented significantly higher apoptosis rates (Q2+Q3)(sh-HERV6196 vs. sh-NC: 27.75% vs. 14.08%) (Fig. 3D) [page 10, lines 181-187]. Figure 3D shows the

overall comparison of apoptosis cells between the sh-NC and sh-HERV6196 groups ($p < 0.05$). The above parts are supplemented in the manuscript.

8. Please label Fig. 7 within the Figures (like labeling parts of an image). What are these cells and which cell lines are used? Fig. 7i-7k lacks convincing evidence. Is this an invasion or migration assay? It is a different scientific term. Matrigel was used in the method, but the figure legend indicates migration.

Response:

This part of the figures has been reassembled and are Fig. 4 now in the manuscript.

Fig. 4 uses the colorectal cancer cell line HCT116, which has been annotated. Fig. 4C (originally Fig. 7i-7k) shows the transwell invasion assay. And Fig. 4C (originally Fig. 7k) show that the migratory ability is reduced after the knockout of HERV6196 compared to the control group ($p < 0.05$).

9. Fig. 10 is not convincing. The ECL method for western is not quantitative. Is there any statistical analysis? What makes this significant?

Response:

In order to further verify the accuracy of the results, we used imageJ to quantitatively analyze the gray scale of the bands. The results showed that after HERV6196 knockdown, the expression of *NEK2* and other genes was decreased, and these genes were related to the pathogenesis and progression of rectal cancer in previous studies. The expression of HERV gene transcripts may be involved in the rectum by regulating these genes The occurrence and development of cancer. This result is shown in a bar chart in the manuscript (Fig. 6A-B) [page 17, line 263-266].

10. Material methods: Please specify sources for all reagents. What is the original source of cells used for this study? What does it mean if this comes from laboratory stocks?

Response:

The sources of materials involved in the experiment have been supplemented. The cells used in this experiment were purchased from Wuhan Pricella Biotechnology Co., Ltd., China [page 23, lines 420-422].

11. The authors downloaded ChIP-seq data from GSE166254. Why were only three samples of tumors and adjacent tissues analyzed? There are more samples and they should be included. Are these ChIP data from samples with upregulated HERV6169? To make it easier for readers to understand, please give an explanation of the ChIP condition and the reason why you used these data. This is the data from ChIP-seq with histone H3 acetylation of Lysine 27, which is a marker for active gene regulation. And please explain briefly why you used ChIP and ATAC-seq data and compared.

Response:

GSE166254 includes fifteen CRC samples and fourteen normal colonic mucosa samples. Since the replicates meet the requirement of more than three samples, we only displayed three CRC samples and three normal colonic mucosa samples. We modified the figure to show fifteen CRC samples in the main text (Fig.5A) [page 15, line 237], and the results of fourteen normal colonic mucosa samples are presented in the supplementary materials (Fig.S1).

The data are not from samples with upregulated HERV6169, but rather from ChIP-seq of H3K27ac conducted on human colorectal cancer samples and normal colonic mucosa. We downloaded the hg19 normalized bw files from GSE166254, and used CrossMap to convert the hg19 normalized bw files to hg38 normalized bw files, which were then exported to IGV for display [page 26, lines 484-489]. Through the Genehancer enhancer database, we found that our HERV6196 has potential enhancer activity (GH01J221965 and GH01J221969). Therefore, we further utilized ChIP-seq data from CRC tissues and discovered the activation of H3K27ac in the HERV6196 region, further confirming the enhancer function of HERV6196 in CRC. A dual luciferase assay was conducted on the function of HERV6196 eEnhancer, with the results indicating that enhancer activity of HERV6196. This result is presented in the results section of the manuscript [page 16, lines 250-255].

We utilized ATAC-seq data to demonstrate the open chromatin accessibility of the HERV6196 region, indicating active epigenetic characteristics of HERV6196. Through a combined analysis of ATAC-seq and ChIP-seq, we confirmed the potential enhancer activity of HERV6196.

12. Too many figures. Figs 1-2, Fig.3-5, and Fig. 8-10 can be combined.

Response:

Based on your suggestion, we combined Figs 1-2, Fig.3-5, and Fig. 8-10.

Re: Spectrum00788-25R1 (**HERV6196 as an enhancer with oncogenic potential in rectal cancer**)

Dear Dr. Jun-Xin Wu:

I am pleased to inform you that your manuscript has been editorially accepted for publication. However, there are a few additional minor questions in the submission form that need to be answered before the final decision. Once these are completed, please return your submission so that I can move your paper forward to acceptance.

Revision Guidelines

Sincerely,
Hyun Jin Kwun
Editor
Microbiology Spectrum

Reviewer #1 (Comments for the Author):

The study reuses public datasets (RNA-seq: GSE50760, GSE104836, GSE142279; ChIP-seq: GSE166254; ATAC-seq from TCGA). Newly generated data appear to be cell-based assays, RT-qPCR, ddPCR counts, and westerns; these are typically provided as Supplementary Files rather than deposited to a raw-reads repository. While might not be necessary, I recommend depositing processed numerical data (e.g., ddPCR copy numbers, RT-qPCR $\Delta\text{Ct}/\Delta\Delta\text{Ct}$, luciferase ratios, colony/scratch/invasion quantifications, western densitometry tables) to a general repository (e.g., Figshare/Zenodo) and citing a DOI, as well as depositing any raw data.

Reviewer #2 (Comments for the Author):

Most of my comments were addressed.

- Delete "Sea Kidney" in line 477.

- How was the "HERV6196 enhancer" engineered into the pGL4 vector? Information for the enhancer sequences and primers used for cloning should be added in the supplementary data.

AMERICAN
SOCIETY FOR
MICROBIOLOGY
Microbiology Spectrum

ASM eJP Peer
Review SystemHome Help for Authors Help for Reviewers Contact Us Logout

Review saved.

You may print this page for your records.

Reviewer's Overall Assessment

Editor	Hyun Jin Kwun
Date Due	October 18, 2025
Manuscript #	Spectrum00788-25R1
Title	HERV6196 as an enhancer with oncogenic potential in rectal cancer
Corresponding Author	Jun-Xin Wu
Contributing Author	Yi-Xiu Gan, Xin Jiang, Zhi-Yu Wang, Yi-Lin Yu, Ling-Dong Shao, Jianmin Wang

Evaluation	
Question	Answer
Recommendation (Required)	Accept
Are the authors' conclusions supported by the data?	Yes
Is manuscript written in comprehensible english?	Yes
Does the study require a public repository?	No
Public repository details (Required)	
Biosafety/Biosecurity Concerns?	No
Statistical tests applied?	Yes
Comments for the Author	The study reuses public datasets (RNA-seq: GSE50760, GSE104836, GSE142279; ChIP-seq: GSE166254; ATAC-seq from TCGA). Newly generated data appear to be cell-based assays, RT-qPCR, ddPCR counts, and westerns; these are typically provided as Supplementary Files rather than deposited to a raw-reads repository. While might not be necessary, I recommend depositing processed numerical data (e.g., ddPCR copy numbers, RT-qPCR $\Delta C_t/\Delta\Delta C_t$, luciferase ratios, colony/scratch/invasion quantifications, western densitometry tables) to a general repository (e.g., Figshare/Zenodo) and citing a DOI, as well as depositing any raw data.
Comments to the Editors (Required)	Are all the authors' conclusions supported by their data? Mostly yes. The manuscript presents consistent evidence that HERV6196 is upregulated in rectal cancer (RT-qPCR across lines and ddPCR in 18 paired tissues, $p=0.0019$), and that knockdown reduces proliferation, alters cell cycle, increases apoptosis, and impairs migration/invasion. It also supports an enhancer role using H3K27ac ChIP-seq/ATAC-seq plus a dual-luciferase reporter, and shows reduced expression of neighboring oncogenic genes at RNA and protein levels after knockdown. Minor remaining caveat is the modest clinical cohort size; the authors acknowledge this and outline plans for expansion. Is the manuscript written in standard English and easy to comprehend? Yes. The prose is clear and professional throughout; figure references and methods are understandable. Minor stylistic tightening could be made, but the language is acceptable for publication.
Reviewer Web of Science (Required)	Yes
Reveal Identity?	Yes

Response to Reviewers

We sincerely thank the editor and the reviewers for their positive feedback and constructive comments on our manuscript. We have addressed all the remaining points raised, and we believe the manuscript has been further improved. Our point-by-point responses are detailed below.

Response to Reviewer #1:

Comment: “The study reuses public datasets (RNA-seq: GSE50760, GSE104836, GSE142279; ChIP-seq: GSE166254; ATAC-seq from TCGA). Newly generated data appear to be cell-based assays, RT-qPCR, ddPCR counts, and westerns; these are typically provided as Supplementary Files rather than deposited to a raw-reads repository. While might not be necessary, I recommend depositing processed numerical data (e.g., ddPCR copy numbers, RT-qPCR $\Delta Ct / \Delta \Delta Ct$, luciferase ratios, colony/scratch/invasion quantifications, western densitometry tables) to a general repository (e.g., Figshare/Zenodo) and citing a DOI, as well as depositing any raw data.”

Response:

We are grateful to the reviewer for this valuable suggestion, which enhances the transparency and reproducibility of our work. In accordance with this recommendation, we have deposited all key processed numerical data underlying the quantitative figures in this study to the Figshare repository.

Dataset Title: Dataset for: HERV6196 as an enhancer with oncogenic potential in rectal cancer.

DOI: 10.6084/m9.figshare.30390199

Private link: <https://figshare.com/s/fb2791600fcc1532f01b>

This dataset includes the quantitative data for: RT-qPCR and ddPCR, Luciferase reporter assays, Colony formation assays, Scratch wound healing assays, Cell

invasion assays, Quantitative analysis of Western blots.

Response to Reviewer #2:

Comment 1: “Delete 'Sea Kidney' in line 477.”

Response:

We thank the reviewer for their meticulous attention to detail. The term "Sea Kidney" has been deleted from line 477 in the Methods section.

Comment 2: “How was the 'HERV6196 enhancer' engineered into the pGL4 vector? Information for the enhancer sequences and primers used for cloning should be added in the supplementary data.”

Response:

We thank the reviewer for this important question. The HERV6196(chr1:221965963-221971618) enhancer, was cloned into the pGL4.23-GV148 vector downstream of the firefly luciferase gene using a restriction enzyme-based homologous recombination method.

Specifically:

- The enhancer genomic sequence (GRCh38/hg38 assembly, chr1:221965963-221971618) was synthesized and amplified via PCR using specific primers.
- The forward primer (5'-3') contained a KpnI restriction site and homologous arm:
TTTCTCTATCGATAGGTACCgtgacttgccccagatggcctgaagtaactgaagaatcacaaaagaa
gtgaaagaccctgcccgaccttaactgatgacattccaccattgtgattgttctgccccaccttaactgagtg.
- The reverse primer (5'-3') contained an XhoI restriction site and homologous arm:
TGCAGATCGCAGATCTCGAGtgaagagaccaccaaacaggctttg.
- The purified PCR product and the KpnI/XhoI-linearized pGL4.23-GV148 vector were assembled using the ClonExpress™ II One Step Cloning Kit, followed by transformation into competent E. coli.

- Positive clones were identified by colony PCR and the correct insertion was confirmed by Sanger sequencing using multiple primers covering the entire enhancer sequence.

As suggested, the detailed enhancer sequence and the full primer sequences have been included in the Supplementary Data.

We once again extend our sincere appreciation to the editor and the reviewers for their time and insightful comments. We hope that our revisions and responses are now satisfactory and that the manuscript is deemed suitable for publication in *Microbiology Spectrum*.

Sincerely,

Jun-Xin Wu (Corresponding Author)

Re: Spectrum00788-25R2 (**HERV6196 as an enhancer with oncogenic potential in rectal cancer**)

Dear Dr. Jun-Xin Wu:

Your manuscript has been accepted, and I am forwarding it to the ASM production staff for publication. Your paper will first be checked to make sure all elements meet the technical requirements. ASM staff will contact you if anything needs to be revised before copyediting and production can begin. Otherwise, you will be notified when your proofs are ready to be viewed.

Sincerely,
Hyun Jin Kwun
Editor
Microbiology Spectrum